# Catalysis of the electrochemical oxygen reduction reaction (ORR) by animal and human cells

**Simon Guette-Marquet[1], Christine Roques[1], Alain Bergel[2]\***

**1** Laboratoire de Génie Chimique, Université de Toulouse, CNRS, INPT, UPS, Fac. Sci. Pharmaceutique, 31062, Toulouse, France, **2** Laboratoire de Génie Chimique, Université de Toulouse, CNRS, INPT, UPS, 31432, Toulouse, France

\* alain.bergel@toulouse-inp.fr

**Data Availability Statement:** All relevant data are within the manuscript.

**Funding:** This work is part of the TECH project: "Looking for extracellular electron transfers with human cells" funded by the French Agence

## Abstract

Animal cells from the Vero lineage and MRC5 human cells were checked for their capacity to catalyse the electrochemical oxygen reduction reaction (ORR). The Vero cells needed 72 hours' incubation to induce ORR catalysis. The cyclic voltammetry curves were clearly modified by the presence of the cells with a shift of ORR of 50 mV towards positive potentials and the appearance of a limiting current (59 µA.cm$^{-2}$). The MRC5 cells induced considerable ORR catalysis after only 4 h of incubation with a potential shift of 110 mV but with large experimental deviation. A longer incubation time, of 24 h, made the results more reproducible with a potential shift of 90 mV. The presence of carbon nanotubes on the electrode surface or pre-treatment with foetal bovine serum or poly-D-lysine did not change the results. These data are the first demonstrations of the capability of animal and human cells to catalyse electrochemical ORR. The discussion of the possible mechanisms suggests that these pioneering observations could pave the way for electrochemical biosensors able to characterize the protective system of cells against oxidative stress and its sensitivity to external agents.

## Introduction

Dissolved oxygen tends to reduce spontaneously on contact with the surface of conductive materials. This spontaneous oxygen reduction reaction (ORR) is very slow at the surface of common carbon-based materials and non-noble metallic materials. However, although slow, it can still be an effective driver of the corrosion of metallic materials [1]. More than 40 years ago, it was discovered that some microbial biofilms have the capacity to catalyse ORR on metallic surfaces and, consequently, to enhance corrosion risks [2]. Since then, many studies have aimed to decipher the mechanisms of ORR microbial catalysis [3]. In 2005, the topic saw a strong revival due to the discovery that ORR microbial catalysis can be exploited to design the cathodes of fuel cells [4,5]. In this context, many studies have demonstrated the capacity of various microbial biofilms to catalyse electrochemical ORR in very efficient ways [3,6]. Two different types of microbial ORR catalysis can be distinguished, as detailed below.

Nationale de la Recherche Scientifique (ANR-17-CE07-45).

**Competing interests:** The authors have declared that no competing interests exist.

On the one hand, an efficient catalytic mode has been pointed out in the many studies dealing with microbial fuel cells [7]. In this framework, ORR is most often achieved by multispecies microbial biofilms formed directly in natural media (seawater [4,8–11], wastewater [12], etc.), or in synthetic media inoculated with natural multi-species inocula [13–19]. Considerable current densities, of the order of an $A/m^2$, and high half-wave potentials, above 0.4 V/SHE, have been reached [14,15,17,19]. The efficiency of the ORR catalysis has been attributed to the enrichment of the biofilm population in a specific bacterial phylum [20,21], class [14,22], family [19] or genus [23,24]. Nevertheless, the strains isolated from these multispecies biofilms lose most of their catalytic efficiency when used under pure cultures [10,25,26]. It is consequently suspected that the efficiency of this type of ORR microbial catalysis comes from synergetic effects that occur inside multispecies biofilms [10], including complex couplings that may involve sulphur [24] or nitric [27] compounds. To the best of our knowledge, efficient ORR microbial catalysis with a pure strain has only been obtained in a very specific case, which took advantage of the capacity of *Acidithiobacillus ferrooxidans* to work in very acidic conditions (pH 2.0) [28,29].

On the other hand, a myriad of bacterial strains have revealed a capacity to catalyse ORR in pure culture conditions [10,25,26,30–35]. Nevertheless, this catalysis is poorly effective and requires transient electrochemical techniques, such as cyclic voltammetry, to be detected. It is generally characterized by an increase in the ORR half-wave potential of only a few tens of millivolts with respect to the wave recorded on clean control electrodes.

In comparison to the numerous reports devoted to bacterial cells, very few reports have investigated ORR catalysis by eukaryotic cells, especially animal or human cells so far. Several studies have dealt with extracellular electron transfer between yeasts and electrodes [36], with or without redox mediators [37]. Various metabolic pathways, such as glycolysis and fermentation, and including aerobic respiration, have been considered as the source of the electrons transferred to the anode [38], but ORR catalysis has not been tackled. Electron transfer with mitochondria from various sources, including yeast and bovine sources, have also been investigated [39,40], without evoking possible ORR catalysis. To the best of our knowledge, only one study has approached this topic, by using red blood cells [41]. Erythrocytes are anucleated cells, daily produced, able to show a great capacity to catalyse ORR from potentials of about 0.05 V/SHE. This catalysis is linked to the specificity of red blood cells, containing high amount of haems known for their capacity to catalyse ORR [33,42].

The present study is a first approach, checking the capacity of animal and human cells to catalyse electrochemical ORR. The animal cell lineage Vero and the human cell lineage MRC5 were used. The Vero cell lineage was established from kidney tissue of a green monkey (*Cercopithecus aethiops*) [43]. MRC-5 is a fibroblastic cell lineage derived from human foetal lung [44]. These two lineages were chosen because they are commonly used in biomedical industries to culture viruses and produce vaccines, and for diagnostic and research purposes [45,46]. Here, the objective was to determine whether such animal and human cells can catalyse electrochemical ORR and to suggest possible applications of this catalysis in the biomedical field.

## Materials and methods

### Electrodes and electrochemical reactors

Screen printed electrodes were purchased from Metrohm-DropSens. These disposable electrodes consist of a plastic strip on which a three-electrode system is screen-printed. The working electrode is a central disk 4 mm in diameter with a circular auxiliary electrode around it (Fig 1). Both electrodes are made of the same material. Two electrode materials were used here: carbon and carbon coated with carbon nanotubes (ref. DRP-110 and DRP-110 CNT).

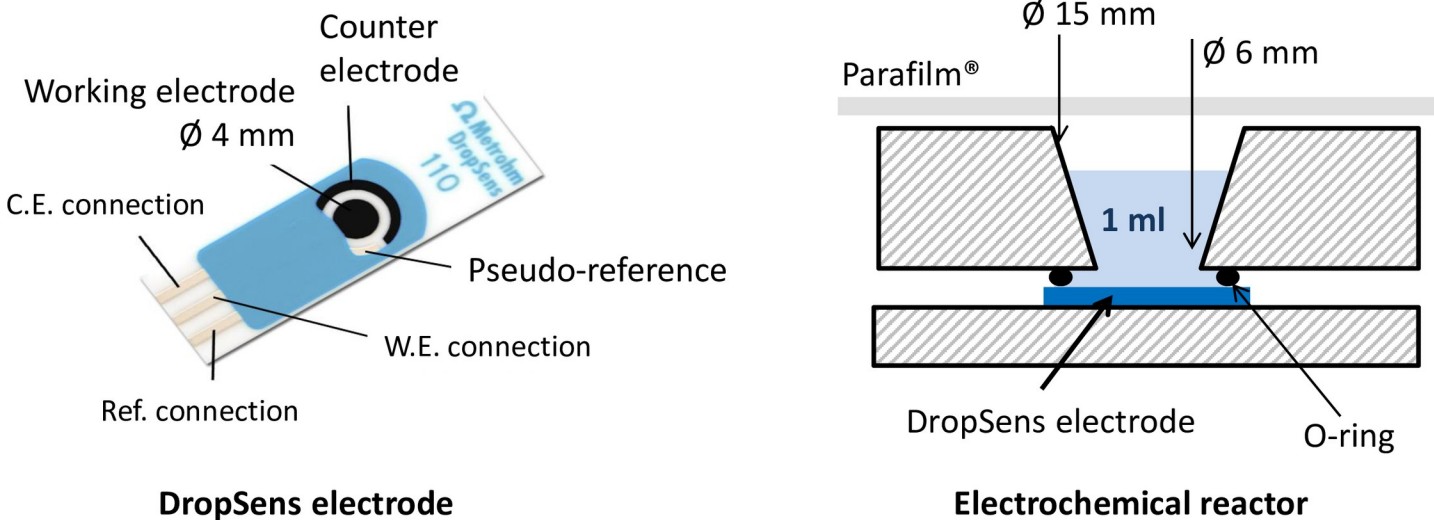

**Fig 1. Schematic of the electrochemical set-up.** Picture of the electrode is adapted from the Metrohm-DropSens commercial documentation.

The reference electrode was a silver pseudo-reference. Its potential, measured in the condition of the present study in the complete growth medium, was 0.049 V with respect to a saturated calomel reference. In the experimental conditions used here, the potential of the DropSens pseudo-reference was consequently equal to 0.290 V/SHE. All the potential values given in the study are expressed with respect to this DropSens pseudo-reference. The potentials can be expressed relative to the standard hydrogen electrode (SHE) by adding 0.290 mV to the values given in the text.

All experiments were performed in commercial reactors suited for the use of electrode strips (CFLWCL-Conic Metrohm-DropSens). In these devices, the electrode strip is maintained at the bottom of the reactor, with the solution over it (Fig 1). Prior to use, electrodes and reactors were sterilized by soaking them in 70% ethanol for 30 minutes. After sterilization, the electrodes were washed three times with ultrapure water. Reactors and electrodes were air dried for 2 hours in a microbiological safety cabinet.

When indicated, electrodes were pre-treated with inactivated foetal bovine serum (FBS) or with poly-D-Lysine (PDL) (Sigma Aldrich). The FBS pre-treatment was carried out directly in the reactors. Sterile electrodes were mounted in the reactor. Reactors were filled with 1 ml FBS diluted by half with PBS. After 2 hours of incubation in sterile safety cabinet, reactors were rinsed with 1 ml of complete medium and immediately seeded with cells according to the procedure described below. For coating with PDL, sterile electrodes were dipped in high molecular weight PDL solution at 100 μg/ml for 2 hours at room temperature, then washed three times with ultrapure water and air dried before use.

## Cell culture

Cells of the Vero cell line (CCL-171 from ATCC) and MRC-5 cell line (CCL-81 from ATCC), kindly provided by Fonderephar (Toulouse, France), were cultured in RPMI growth medium (Gibco) supplemented with 10% heat inactivated foetal bovine serum (FBS), 2 mM of L-glutamine (Gibco) and antibiotics (penicillin 10 U/ml and streptomycin 10 μg/ml, from Gibco), at 37°C in humidified air with 5% $CO_2$. Cells were routinely subcultured at 80–90% confluence.

### Reactor seeding

Cells from a 75 cm$^2$ flask at 70–90% confluence were harvested with trypsin-based buffer. After centrifugation at 200 g for 5 minutes, the cell pellet was resuspended in complete growth medium (RPMI, 10% FBS, L-glutamine, antibiotics) at a density of 1 000 000 cells/ml. Electrochemical reactors were seeded with a small volume of cell suspension in order to obtain the required cell concentration. The final volume of the reactors was adjusted to one ml by addition of sterile complete medium. Reactors were closed with Parafilm® to ensure sterility and immediately transferred to a 37˚C/5% $CO_2$ incubator for the indicated incubation time.

### Electrochemical recording and analysis

Cyclic voltammograms (CVs) were recorded using a VMP-3 potentiostat (Bio-Logic, France) monitored by the EC lab software. In each experimental set, several reactors were run in parallel, always with one or two control reactors without cells. The values of all the electroanalytical parameters were constant. The potential scan rate was 10 mV.s$^{-1}$. The minimum and maximum limits of the potential scan were always -0.2 V and 0.5 V, respectively for the Vero cells and -0.4 V and 0.6 V for the MRC-5 cells. Each CV was recorded three times. The first cycle was sometimes slightly different from the others but the second and third cycles were identical. The second cycles are presented here. Measurements were made at room temperature (22˚C ± 2˚C). For supernatant assays, medium collected from the reactors was centrifuged at 1000 g for 10 min. Supernatants were then analysed by CV as described above.

The effectiveness of the catalytic effect was assessed by measuring the shift towards positive potential values (ΔE) provoked by the presence of the cells at a given value of the current. At the chosen value of the current, ΔE was measured as the difference between the potential of the CV recorded in the presence of cells minus the value of the control CV without cells. When two controls without cells were performed in the same set of experiments, the potential shift was calculated with respect to the average value of the potential of the controls. ΔE was measured for the current of 5 µA for the Vero cells (Table 1) and the MRC5 cells incubated for 24 hours (Table 3) because it was the region presenting the most marked potential shifts for most of the reactors. It was measured at 3 µA for the MRC5 cells incubated 4 hours (Table 2) because the currents were lower in these experiments.

The value of the limiting current ($I_{lim}$) observed on the CVs was assessed by the value of the current at the lower limit of the potential scan (-0.2 V/ref for the Vero cells and -0.4 V/ref for the MRC5 cells) minus the value of the capacitive current, which was measured at 0.2 V/ref.

### Microscopy

After CV recording, the medium in the reactors was discarded. The reactors with the electrode strip inside were washed with 500 µl PBS pre-warmed to 37˚C, then filled with 500 µl of staining solution (5 µM Syto9® and 1 mg/ml propidium iodine from Molecular Probes Inc., in PBS) incubated at 37˚C for 15–30 min. After incubation, the electrodes were removed from the reactors and transferred in Petri dishes filled with 15 ml of PBS. Images were acquired with an Axiotech epi-fluorescence microscope (Zeiss) equipped with green and red fluorescence filters (41001HQ F C71828 and 41005HQ PI C71829 from Zeiss). Images were processed using Zen (blue edition) software (Zeiss, release 2.5).

## Results

### Vero cells

Reactors equipped with carbon electrodes were inoculated with 30 000 cells and incubated for 24, 48 and 72 h. Each set of experiments was carried out with three or four reactors in parallel,

**Table 1. Analysis of the reactors incubated for 72 hours with Vero cells.**

| Number of cells | Electrode | Control or cells | $|I_{lim}|$[1] $(\mu A)$ | Potential (E) at 5 µA $(V/ref)$ | Shift vs. control [2] ΔE at 5 µA $(mV)$ |
|---|---|---|---|---|---|
| 30 000 | C | Control | - | -0.151 | - |
| | | Control | - | -0.139 | - |
| | | Cells | 6.9 | -0.085 | 60 |
| | | Cells | 4.3 | -0.075 | 70 |
| | CNT | Control | - | -0.132 | - |
| | | Control | - | -0.132 | - |
| | | Cells | 5.2 | -0.078 | 54 |
| | | Cells | 5.1 | -0.073 | 59 |
| | CNT | Control | - | -0.121 | - |
| | | Control | - | -0.128 | - |
| | | Cells | 11.8 | -0.096 | 28 |
| | | Cells | 7.2 | -0.082 | 42 |
| 100 000 | CNT | Control | - | -0.126 | - |
| | | Control | - | -0.113 | - |
| | | Cells | 5.5 | -0.075 | 44 |
| | | Cells | 5.4 | -0.038 | 81 |
| 40 000 | CNT-FBS | Control | - | -0.127 | - |
| | | Control | - | -0.105 | - |
| | | Cells | 8.1 | -0.085 | 31 |
| | | Cells | 9.8 | -0.068 | 48 |
| 15 000 | C-FBS | Control | - | -0.130 | - |
| | | Cells | 9.6 | -0.085 | 45 |
| | | Cells | 8.3 | -0.075 | 55 |
| | C-FBS | Control | - | -0.127 | - |
| | | Cells | 7.8 | -0.081 | 46 |
| | | Cells | 8.2 | -0.081 | 46 |
| | C-FBS | Control | - | -0.136 | - |
| | | Cells | 8.0 | -0.097 | 39 |
| | | Cells | 8.0 | -0.082 | 54 |

C: Carbon electrode; CNT: Carbon nanotube-coated electrode; FBS: Foetal bovine serum.

[1] $I_{lim}$ is the value of the current at the lower potential limit of the CV records (-0.2 V/ref) minus the capacitive current measured at 0.2V/ref.

[2] When the experimental set-up had two control reactors, the potential shift is calculated with respect to the average value of the potential of the two controls.

systematically including two reactors inoculated with cells and one or two control reactors without cells. Cyclic voltammograms (CVs) recorded at the end of the incubation periods of 24 h showed no difference in the presence or absence of cells. After 48 h of incubation, two of the four reactors inoculated with cells, showed a very weak catalytic effect, while the others gave in the presence of cells identical or lower current than without cells (Fig 2A).

In contrast, after 72 h of incubation, all CV records showed clear changes in the area of ORR (Fig 2B). On the one hand, the ORR wave was displaced towards higher potentials, which denoted a catalytic effect of the cells towards electrochemical ORR. On the other hand, a limiting current appeared on the reduction current, which denoted the appearance of a rate-limiting step due to the cells.

Similar observations were made when using carbon electrodes coated with carbon nanotubes (CNT) (Fig 3). As observed with simple carbon electrodes, no significant change was provoked by the cells during 48 h of incubation, while clear ORR catalysis was observed after

**Table 2. Analysis of the reactors incubated for 4 hours with MRC5 cells.**

| Number of cells | Electrode | CV shape and $E_{peak}$ (V/SCE) | Potential with cells E at 3 μA (V/ref) | Shift vs. control ΔE at 3 μA (mV) |
|---|---|---|---|---|
| 50 000 | C | No catalytic effect | | |
| | CNT | No catalytic effect | | |
| | C | Peak | -0.106 | 60 |
| | CNT | Peak | -0.112 | 62 |
| 200 000 | C-PDL | Peak | -0.111 | nd [1] |
| | CNT-PDL | Peak | -0.066 | 287 |
| | C-PDL | Peak | -0.072 | 120 |
| | CNT-PDL | Peak | -0.078 (at 1.5 μA) [2] | 110 (at 1.5 μA) [2] |
| | C | Shift | -0.134 | 137 |
| | CNT | Peak | -0.143 | 87 |
| | C | Peak | -0.106 | 154 |
| | CNT | Peak | -0.162 (at 5mVs$^{-1}$) [3] | 145 (at 5 mVs$^{-1}$) [3] |
| | CNT | Peak | -0.114 | 166 [4] |
| | | Shift | -0.231 | 49 [4] |
| | CNT-PDL | Peak | -0.125 | 115 [5] |
| | | Peak | -0.202 | 38 [5] |
| | C | Peak | -0.119 | 87 |
| | | Peak | -0.169 | 37 [6] |
| | C | Shift | -0.130 | 127 [5] |
| | | Shift | -0.219 | 38 [5] |

C: Carbon electrode; CNT: Carbon nanotube-coated electrode; PDL: Poly-D-lysine.

[1] the control curve did not reach 3 μA.

[2] the curve did not reach 3 μA.

[3] the curves recorded at 10 mv.s$^{-1}$ could not be exploited because of electrical interferences.

[4] the two controls without cells were slightly different, the average E value at 3 μA was -0.280 V.

[5] two perfectly identical controls.

72 h of incubation. Increasing the number of cells used to inoculate the reactor from 30 000 to 100 000 led to the same catalytic effect after 72 h of incubation, without enhancing it.

Three other sets of experiments were performed by inoculating the reactors with 15 000 cells. The incubation time was always 72 h. The carbon electrodes were first treated by the adsorption of diluted foetal bovine serum (FBS), with the intention of promoting cell adhesion on the electrode surface. The three sets displayed similar catalytic effects as previously.

Finally, a similar set of experiments (inoculation with 15 000 cells) was performed in the absence of oxygen (84–85% $N_2$, 5% $CO_2$ and 0–1% $O_2$) during the 72 h of incubation. The CVs recorded in the control reactor without cells and in the inoculated reactors were identical. None showed any reduction wave (Fig 4), which confirms that the reduction wave was due to ORR.

The effectiveness of the catalytic effect was assessed by measuring the shift towards positive potential values provoked by the presence of the cells at a given value of the current (ΔE). The potential shifts resulting from the 16 electrodes incubated for 72 h was 50.1 ±13.5 mV (16 experiments with cells, 13 controls w/o cells, Table 1). The catalytic effectiveness was independent of the nature of the electrode (carbon or CNT-coated carbon), of the initial cell number in the range 15 000 to 100 000, and of the electrode pre-treatment by FBS adsorption.

After 72 h of incubation, the cells provoked appearance of a limiting current ($I_{lim}$) at the lowest potential values, which was not observed on the control experiments without cells. The

**Table 3. Analysis of the reactors incubated for 24 hours with MRC5 cells.**

| Number of cells | Electrode | Control, supernatant or cells | $|I_{lim}|$[1] ($\mu A$) | Potential (E) at 5 μA (V/ref) | Shift vs. control ΔE at 5 μA (mV) |
|---|---|---|---|---|---|
| 50 000 | C | control | 8.6 | -0.152 | - |
| | | supernatant | 8.0 | -0.125 | 27 |
| | | cells | 6.6 | -0.065 | 87 |
| | CNT | control | 8.0 | -0.099 | - |
| | | supernatant | 8.7 | -0.081 | 18 |
| | | cells | 6.0 | -0.060 | 39 |
| | C | control | 8.5 | -0.154 | - |
| | | supernatant | 7.9 | -0.098 | 56 |
| | | cells | 5.5 | -0.065 | 89 |
| | CNT | control | 8.4 | -0.205 | - |
| | | supernatant | 9.1 | -0.194 | 11 |
| | | cells | 5.3 | -0.061 | 144 |
| 200 000 | C | control | 8.9 | -0.141 | - |
| | | supernatant | 8.1 | -0.065 | 76 |
| | | cells | 2.9 | nd [2] | nd [2] |
| | CNT | control | 8.0 | -0.110 | - |
| | | supernatant | 8.5 | -0.054 | 56 |
| | | cells | 4.7 | -0.088 | 22 |

[1] $I_{lim}$ is the value of the current at the lower potential limit of the CV records (-0.4 V/ref) minus the capacitive current measured at 0.2V/ref.

[2] the curve did not reach 5 μA.

limiting current was equal to 7.45 ±2.00 μA (16 experiments), i.e. a current density relating to the surface area of the working electrode equal to 59 ±16 μA.cm$^{-2}$.

The value of the limiting current in absolute value ($|I_{lim}|$) changed in relation to the value of the potential shift (ΔE). The general trend suggests that, the greater the potential shift was, the lower the limiting current became (Fig 5). Consequently, the more efficient the catalysis, the greater the limiting step it introduced, which reduced the current produced at the low potential values.

The oxygen reduction reaction:

$$O_2 + 2\,H_2O \rightarrow 4\,OH^- \tag{1}$$

is sensitive to the pH value of the solution. It is enhanced by acidic conditions. The initial value of the pH of the culture medium was 7.2 ±0.1. It decreased slightly, to 6.8, after 72 h of incubation. The possible impact of such an acidification on ORR was assessed by recording control CVs in the complete growth medium without cells after decreasing its pH from 7.2 to 6.7 by adding a small quantity of hydrochloric acid. The pH decrease did not affect the shape of the CV records–in particular, no limiting current appeared. At the value of 5 μA, the ORR current was not shifted by more than a few mV (less than 5 mV) towards positive potentials. Consequently, the weak acidification of the medium due to cell growth cannot be responsible for the ORR catalysis induced by the cells.

After CV recording, epi-fluorescent microscopy was performed to check cell growth on the electrode surface. After 72 h of incubation, the electrode surface was almost completely covered by Vero cells (Fig 6). The cells adhered to the electrode surface and formed a confluent cell layer. Cells on the electrode exhibited two types of film pattern. Some areas showed well individualized cells, while other areas showed cell aggregates where cell outlines were not clearly visible. The coexistence of these two cell mat patterns can be explained by an imaging artefact in the areas

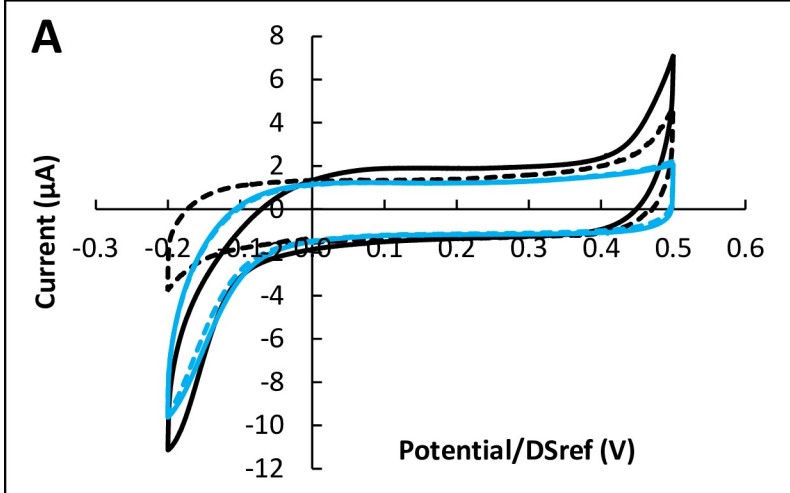

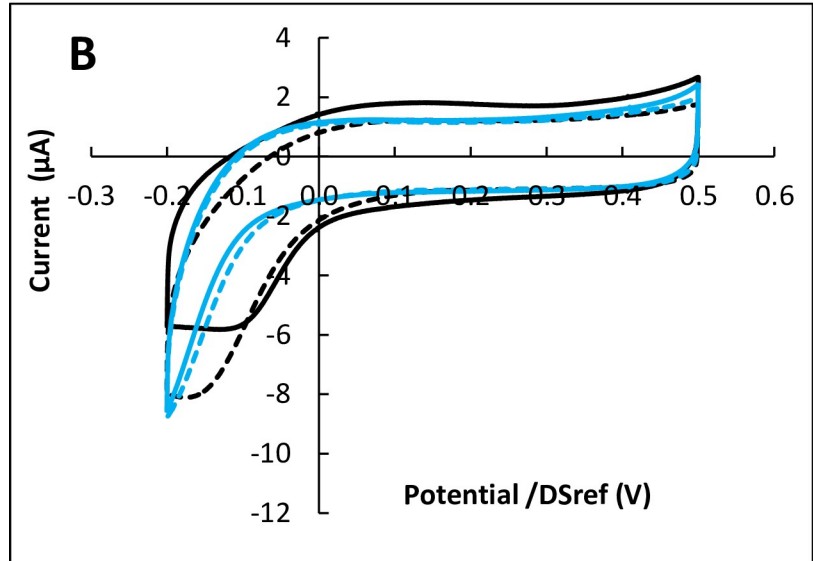

**Fig 2. Cyclic voltammetry of Vero cells on carbon electrodes after 48 and 72 hours of incubation.** Potential scan rate 10 mV.s$^{-1}$. A) 48 hours, B) 72 hours. Black: with cells; blue: controls without cells; in each case dotted lines are duplicates. Potentials are expressed vs. the DropSens pseudo-reference (DSref), values vs. SHE can be obtained by adding 0.290 V.

where cells accumulated in superimposed multi-layers. The 2-dimensional imaging crushed the different superimposed layers, resulting in the impression of fuzzy cell aggregates.

The most important result is the very small number of damaged cells, which appeared stained in red on the epi-fluorescence images. It can be concluded that the cell mat remained viable on the electrode surface after 72 h of incubation. Furthermore, electrochemical CV recording was innocuous for the cells.

## MRC5 cells

Similar experiments were carried out with MRC5 cells. Ten independent experimental set-ups were used, each composed of two reactors inoculated with cells and one or two controls

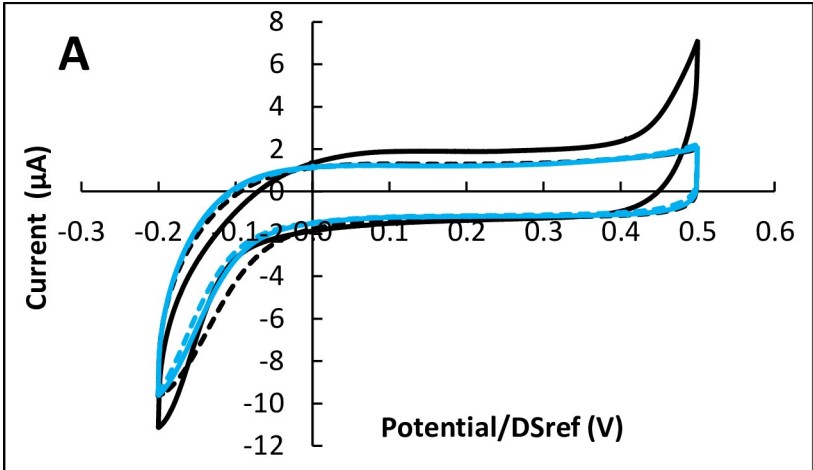

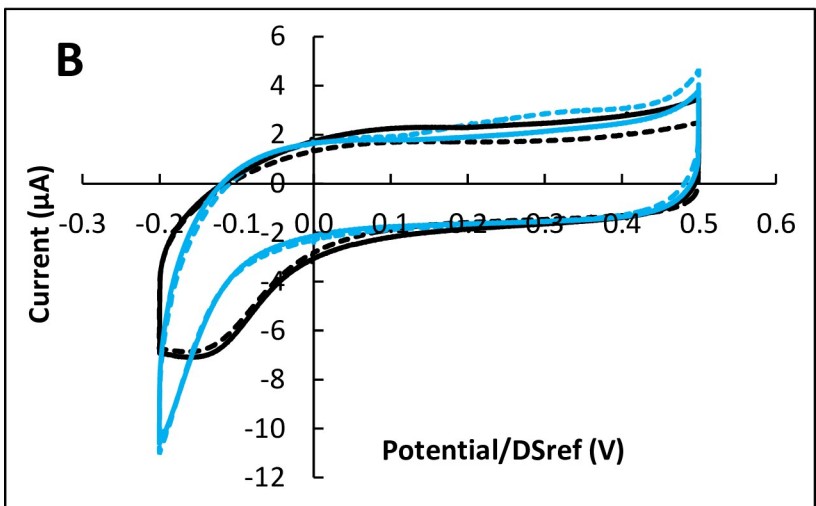

**Fig 3. Cyclic voltammetry of Vero cells on CNT-coated carbon electrodes after 48 and 72 hours of incubation.** Potential scan rate 10 mV.s$^{-1}$. A) 48 hours, B) 72 hours. Black: with cells; blue: controls without cells; in each case dotted lines are duplicates. Potentials are expressed vs. the DropSens pseudo-reference (DSref), values vs. SHE can be obtained by adding 0.290 V.

without cells. In some cases the electrodes were pre-treated with poly-D-lysine (PDL) with the intention of promoting cell adhesion on their surface. PDL is a synthetic, positively charged polymer, which binds to the negatively charged cell membrane through electrostatic interaction and is thus commonly used to promote cell adhesion on solid surfaces [47].

With MRC5 cells, it was not necessary to wait for the ORR catalysis to appear after several days of incubation; ORR catalysis became established after only 4 hours. Reactors inoculated with 50 000 cells showed no catalysis or only a weak effect (Table 2). The inoculation ratio was consequently increased to 200 000 cells, which led to stable results.

The shape of the CV curves was poorly reproducible. The ORR catalysis led to different types of CV shapes as illustrated in Fig 7, which deliberately presents the most extreme cases observed. Some records presented a well-formed peak (with Fig 7A and 7B), while others showed just a more or less significant shift of the abiotic curve towards positive potential values

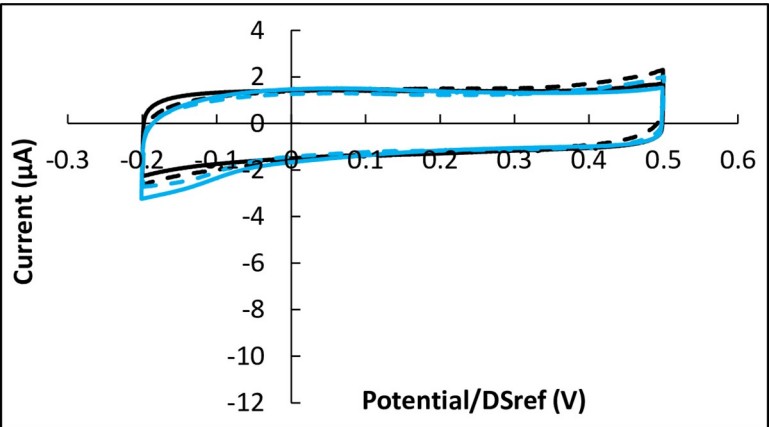

**Fig 4. Cyclic voltammetry of Vero cells on pre-treated carbon electrodes after 72 hours of incubation in the absence of oxygen.** Inoculation with 15 000 cells; potential scan rate 10 mV.s$^{-1}$. Black: with cells; blue: controls without cells; in each case dotted lines are duplicates. Potentials are expressed vs. the DropSens pseudo-reference (DSref), values vs. SHE can be obtained by adding 0.290 V.

(Fig 7C). No correlation could be established between the CV shapes and the electrode material. Surface coating with carbon nanotubes (CNT) and the pre-treatment with poly-D-lysine did not have any obvious effect.

According to the basic principle of CV, two kinds of peak can appear on CV records [48]. It can be roughly summarized that peaks of one kind have a symmetrical shape with the current returning to zero after the peak, while the others are not symmetrical and are followed by a non-zero value plateau, which is the so-called limiting current. In the first case, the peak can be attributed to the electrochemical reaction of species adsorbed on the electrode surface or confined in a thin layer close to the electrode surface. The second case corresponds to the reaction of species coming from the solution. The peak is then due to the transient mass transport limitation that occurs in the diffusion layer when the electrochemical kinetics increases fast.

Here, the current did not fall back to zero after the peak, so the peaks belong to the second type. During a potential scan, the electrochemical rates of oxygen reduction increase so fast

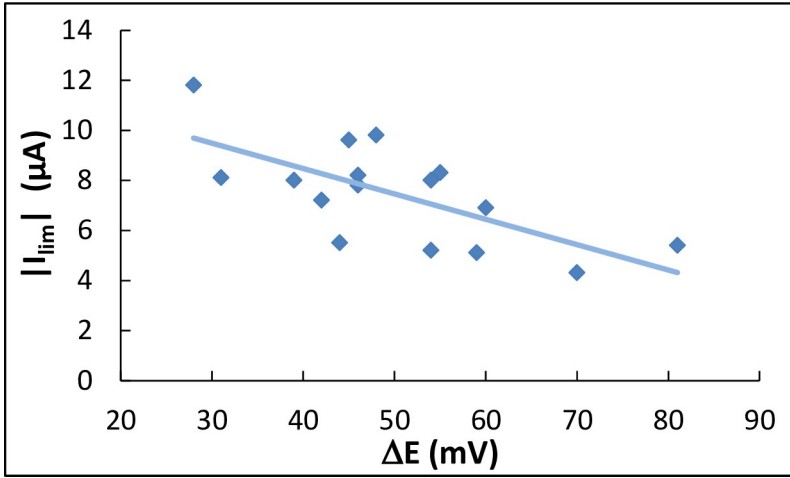

**Fig 5. Limiting current (absolute value) as a function of the potential shift ΔE (from Table 1).** The continuous line is the linear regression.

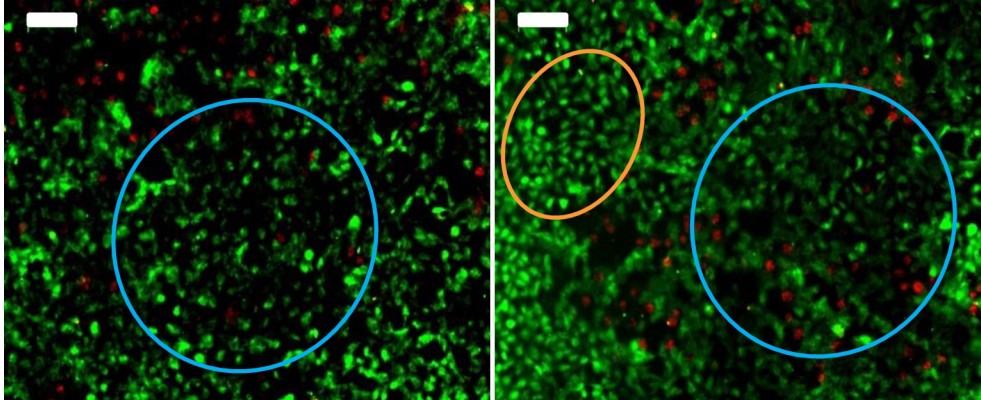

**Fig 6. Epi-fluorescence microscopy images of Vero cells on CNT-coated carbon electrodes (scale bar: 100 μm).**
Reactors were inoculated with 30 000 cells. After 72 hours of incubation electrodes were stained with Syto9 ® (green, viable cells) and IP staining (red, damaged cells). Orange circle: Typical area with individualized cells. Blue circle: Area of aggregate cell layer with blurred boundaries between cells.

that mass transport cannot compensate for the depletion of the diffusion layer. The presence of a peak on CV records (with Fig 7A and 7B) consequently indicates a more efficient catalysis of the electrochemical ORR than the simple shift of the abiotic curve (Fig 7C).

Among the 16 biocathodes designed with 200 000 cells (Table 2), 12 revealed a more or less marked peak and only 4 a shift in potential. This shows the efficiency of MRC5 cells to catalyse electrochemical ORR. The catalytic effect was characterized by measuring the potential shift due to the presence of the cells. Thirteen CV records were thus characterized, resulting in a shift of 110 ±70 mV towards positive potentials.

Actually, the perceived low reproducibility of the CV shape was largely due to the low reproducibility of the control experiments without cells, as can be observed in Fig 7. The culture complete growth medium had a complex chemical composition, including many biochemical compounds, in particular the proteins coming from the foetal bovine serum. There is a concern that some compound(s) may slowly adsorb on the electrode surface and have a weak ORR catalytic effect. Furthermore, the sterilization process of the electrodes, achieved with ethanol, can affect the structure of the carbon (communication from the supplier). Combining the effect of the sterilization step with the subsequent slow adsorption of compounds contained in the culture medium may result in a poorly controlled evolution of the electrode properties with respect to electrochemical ORR. The 4 h incubation may consequently be too short to allow the electrode to reach a stable surface state.

For 9 reactors inoculated with 200 000 cells, the capacity of the supernatant to catalyse ORR was checked. At the end of the 4 h incubation, the supernatant was collected and used to record CV with clean electrodes. In all cases, the supernatant did not show any ORR catalytic effect. On the contrary, it always gave lower reduction current than the control reactors, as illustrated in with Fig 7B and 7C.

After 4 h of incubation and CV recording, microscopy imaging showed only a small number of MRC-5 cells adhering to the surface of the electrodes (Fig 8). In comparison, the density of adhered cells was considerably higher on the surface of the plastic strips. All the cells present on working electrodes exhibited the classical round shape, more typical of non-adherent cells. In contrast, on the plastic strip, some cells started to flatten and spread out, which is characteristic of cells that start to interact with the support. This comparison points out the poor adhesion of MRC-5 cells to the working electrodes. A similar observation was made on CNT-

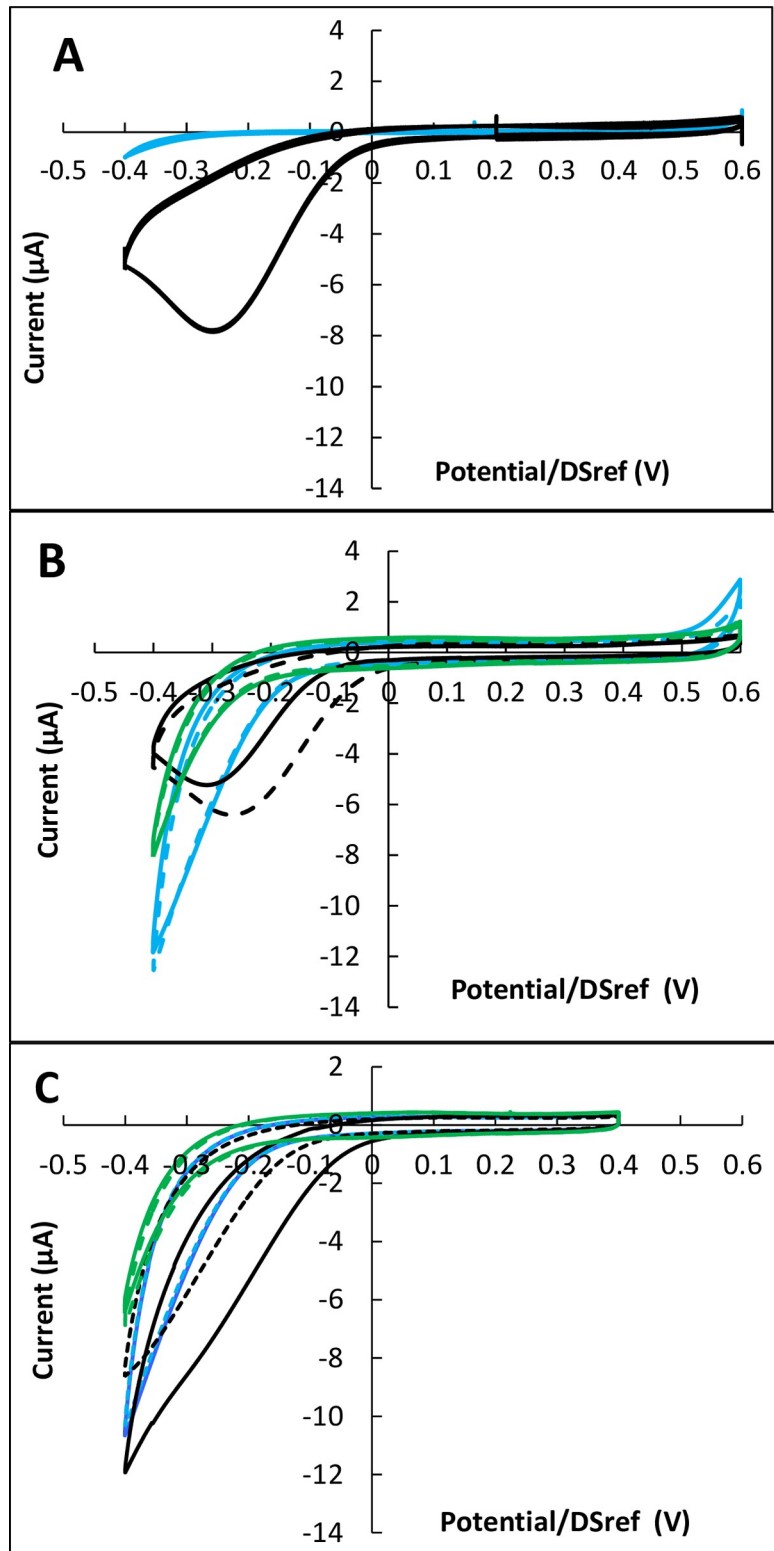

**Fig 7. Cyclic voltammetry of MRC5 after 4 hours' incubation.** Inoculation with 200 000 cells; potential scan rate 10 mV.s$^{-1}$. A) carbon electrodes coated with poly-D-lysine; B) carbon electrodes coated with CNT and poly-D-lysine; C) carbon electrodes. Black: With cells; blue: Control without cells; green: Supernatant; in each case dotted lines are duplicates. Potentials are expressed vs. the DropSens pseudo-reference (DSref), values vs. SHE can be obtained by adding 0.290 V.

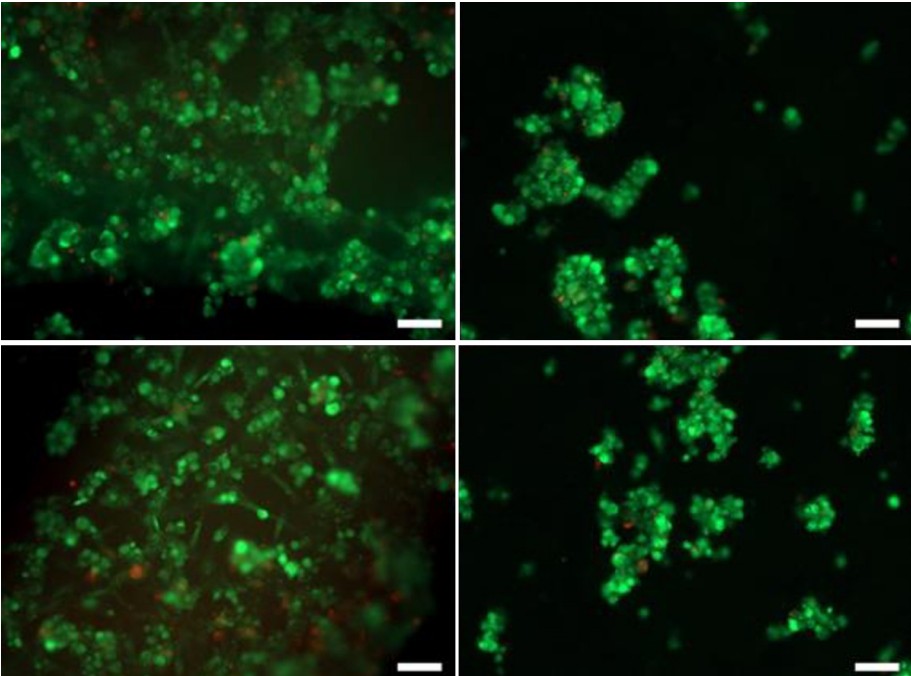

**Fig 8. Epi-fluorescent microscopy images of MRC-5 cells on carbon electrodes pre-treated with poly-D-lysine (scale bar 100 μm).** Reactors were inoculated with 200 000 cells. After 4 hours of incubation, electrodes were stained with Syto9 ® (green, total cells) and IP staining (red, damaged cells). Left column: Plastic strip between the working and auxiliary electrodes; right column: On the working electrode.

coated carbon electrodes and whether or not the pre-treatments with poly-D-lysin or serum were applied did not significantly improve MRC-5 adhesion, regarding the short contact time (no or poor cell replication).

It should be noted that epifluorescence imaging is performed after successive steps of electrode straining and washing, so that only the cells actually adhering to the electrode surface are finally imaged. Consequently, even though microscopy imaging showed only a few adhered cells, it is likely that many more cells were present over the electrode surface during the CV recordings, due to sedimentation. The impact of CV shape was actually due to the presence of cells, whether the cells adhered firmly to the electrode surface or not.

The epifluorescence method makes a cell with a damaged membrane fluoresce in red. Here, very few cells fluoresced in red, which shows full viability of the adhered MRC5 cells.

Three other experimental set-ups, each with two inoculated reactors and two controls without cells, were implemented by lengthening the incubation time to 24 h. The reactors were incubated with 50 000 or 200 000 cells. After 24 h of incubation, all the CV records displayed a reproducible general shape, with a clear ORR wave characterized by a potential shift and a limiting current ($I_{lim}$) (Fig 9).

The control experiments also gave reproducible CV patterns, with ORR waves and close $I_{lim}$ values (8.4 ±0.4 μA, 6 experiments, Table 3). This confirmed that the control electrodes obtained after only 4 h of incubation were in a non-stable phase, still evolving towards higher ORR catalytic effectiveness, probably because of the slow adsorption of compounds from the medium. This could be a major cause of the low reproducibility observed after 4 h of incubation.

The cells provoked a catalytic shift in the range from 39 to 144 mV for the reactors inoculated with 50 000 cells, on average of 90 ±43 mV. The catalytic effect obtained with 50 000 cells

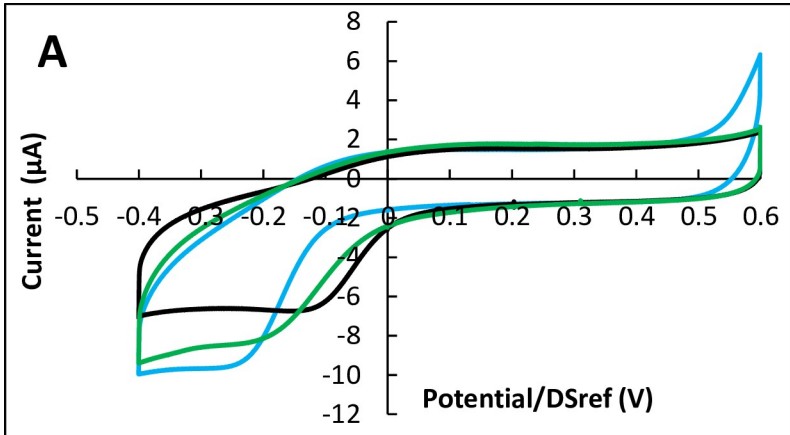

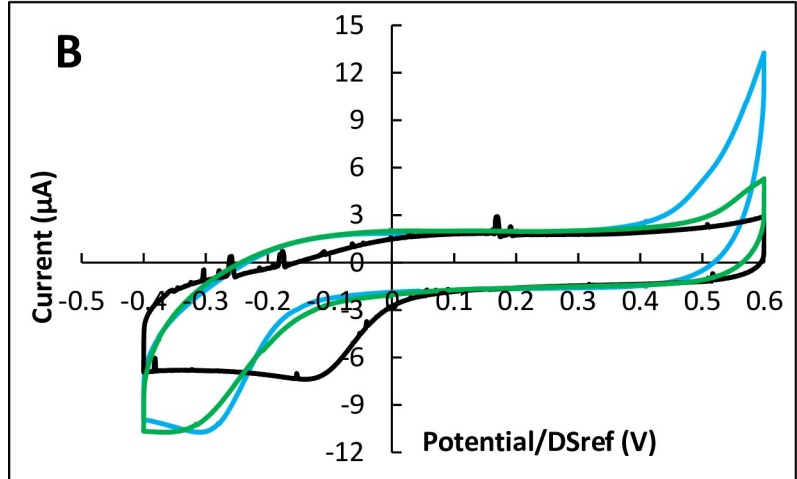

**Fig 9. Cyclic voltammetry of MRC5 after 24 hours' incubation.** Inoculation with 50 000 cells; potential scan rate 10 mV.s$^{-1}$; A) carbon electrodes; B) carbon electrodes coated with CNT. Black: With cells; blue: Control without cells; green: Supernatant; in each case dotted lines are duplicates. Potentials are expressed vs. the DropSens pseudo-reference (DSref), values vs. SHE can be obtained by adding 0.290 V.

after 24 hours of incubation was consequently of the same order of magnitude as that observed after only 4 hours of incubation with 200 000 cells. Here, increasing the inoculation ratio to 200 000 cells did not improve the catalytic effect, but rather seemed to weaken it. Actually, it was difficult to assess the real impact of 200 000 cells on ORR catalysis because high cell concentration considerably decreased the limiting current. $I_{lim}$ was 2.9 and 4.7 µA with 200 000 cells, while it was 5.8 ±0.6 µA with 50 000 cells (4 electrodes) and 8.4 ±0.4 µA for the control without cells (Table 3).

CV recording with clean electrodes in the supernatant showed values of the limiting current ($I_{lim}$ = 8.4 ±0.5 µA) similar to those of the control reactors. The presence of the cells on the electrode surface was consequently the cause of the decrease of the limiting current after 24 h of incubation. In contrast to what was observed after 4 h of incubation, the supernatant induced an ORR catalytic effect here. The effect was more or less significant, with a potential shift in the range of 11 to 76 mV, but it was observed for all six reactors run with the supernatant. It seemed to be more pronounced with 200 000 cells.

## Discussion

The Vero cells showed reproducible ORR catalysis, which required 72 hours of incubation to become established. The MRC5 cells achieved similar catalysis after only 4 hours of incubation. In this case, the weaker reproducibility of the results may have been due largely to the non-stable state of the electrode surface, which was still evolving after 4 h of exposition to the medium. Working on the electrode material and electrode surface preparation to stabilize ORR on the control performed without cells should prove to be a suitable future line of research to improve the results.

A limiting current appeared in the presence of the Vero cells, while it was not observed on the controls recorded without cells. With the MRC5 cells, the limiting current, which was observed on the control records, was reduced by the presence of the cells. It should be noted that the difference between the controls of the Vero cell assays and the controls of the MRC5 cell assay was only due to the difference in the lower potential limits: -0.2 V/ref for the Vero cells and -0.4 V/ref for MRC5 cells. The limiting current was not observed on the controls performed for the Vero cells because the potential was not scanned to a level low enough to record it. The different values of the lower potential limit were fixed on the basis of preliminary experiments seeking to optimize the experimental conditions. In both cases, the presence of the cells diminished the current provided at the lowest potentials, either by making a current limit appear or by reducing its value in comparison to that of the controls.

As the electrodes were at the bottom of the reactors, the cells sedimented and formed a film over the electrode surface. This film contained adhered cells with the Vero cells as can be observed in some spots of the epifluorescence microscopy images (Fig 6), and likely extended during incubation. With the MRC5 cells, the limited film was mainly composed of cells deposited by sedimentation. This film can explain the lowering of the current at the lowest potentials in two ways: the film hindered the mass transport of oxygen and/or the cells consumed oxygen over the electrode surface.

The behaviour of the limiting current ($I_{lim}$), observed with the Vero cells, which decreased (in absolute value) when the catalytic efficiency ($\Delta E$) increased (Fig 5) can thus be explained by the formation of such a film on the electrode surface. The film was more compact, and further impeded the transport or oxygen and/or consumed more oxygen when it was more effective in terms of ORR catalysis. Similarly, the deposited MRC5 cells probably formed a more compact film after 24 h of incubation than after 4 h, explaining why the limiting effect was observed on the CV recorded after 24 h and not detected after only 4 h of incubation.

At the level of this pioneering work, it is difficult to answer the common question of electron transfer mechanisms. In the context of microbial electrochemistry, even after decades of studies, the exact mechanisms are far from having been fully deciphered. Many electron pathways have been speculated, including the most fascinating: direct electron transfer from the cathode to the microbial cells [49]. The cell takes the electrons from the electrode and releases them to oxygen, in a way that enables them to acquire energy [35] (Fig 10A). In the context of ORR catalysis, this pathway has been evidenced with only a few microbial species so far, mainly *Shewanella sp.* [32,35]. Other final electron acceptors such as $CO_2$ [34] and fumarate [50] can also be reduced following this mechanism. The pioneering results described here are not sufficient to establish whether such direct electrode-cell electron transfer occurred or not. Nevertheless, the necessity for the Vero cells to become adhered to the electrode surface may be an element supporting this hypothesis.

Involvement of redox proteins, either bound to the external membrane [30,31] (Fig 10B) or released by the cells onto the electrode surface [26] (Fig 10C), is the assumption that is the most widely made to explain microbial ORR catalysis. Anti-oxidant related enzymes such as

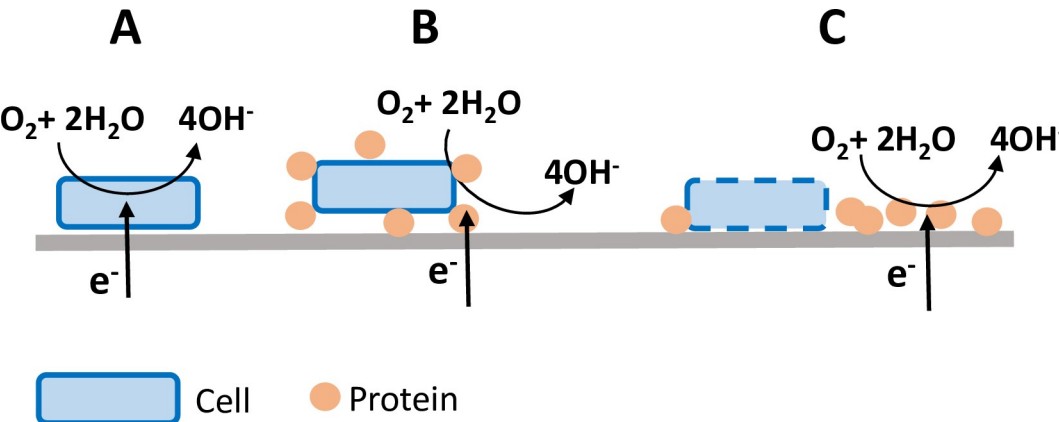

**Fig 10. Three speculated electron transfer pathways.** A) Catalysis by cellular metabolism via direct electron transfer from the electrode to the cell, B) catalysis due to membrane-bound proteins, C) catalysis by proteins released onto the electrode surface.

catalases, superoxide dismutase and peroxidases, which are in charge of protecting the cells against oxidative stress, are suspected to play key roles in these mechanisms [26,30,31,51,52]. On the one hand, these proteins are ubiquitously present in aerobic species. On the other hand, many analytical studies have shown their capacity to catalyse electrochemical ORR, for instance by recycling $H_2O_2$ to $O_2$ [51–53] or by creating an internal electron transfer pathway from the cathode to oxygen [54–58].

Organs and tissues are equipped with various antioxidant systems depending on their metabolic activity and rate of oxygen consumption [59]. MRC5 cells are fibroblasts from lung tissue. They are destined to be exposed to a particularly oxygen-rich environment and should consequently have a very effective pool of antioxidant-related enzymes to protect them against strong oxidative stress. This could explain the MRC5 cells' ability to achieve effective ORR catalysis after only 4 h of incubation, while 72 hours were required for the Vero cells. The different behaviours observed here between MRC5 and Vero cells is consistent with the involvement of antioxidant-related enzymes in the ORR catalysis mechanism.

After 4 h of incubation, the supernatants collected from the culture medium did not show any ORR catalysis. ORR catalysis was consequently due to the action of membrane-bound proteins. In contrast, after 24 h of incubation, the ORR catalysis observed with the supernatants showed that the relevant compounds were released to the medium. These observations were also consistent with a catalytic mechanism that involves proteins, which were still bound to the cell membrane after 4 h of incubation and then released to the medium after 24 hours.

There is a wealth of literature on the role of oxidative stress in diseases such as diabetes [60], neurodegenerative disorders [61], cardiovascular diseases [62] and Alzheimer's troubles [63]. Environmental factors (e.g. exposure to pesticides, organic toxic compounds and (nano) particles) can favour these diseases, and many investigations have shown that differences in the oxidative protective systems of the tissues may be the basis of their different susceptibility to various environmental toxic agents [59].

It is consequently of high interest to develop any tool that could quantify the oxidative protective system of cells and to assess the impact of toxic agents on them or the efficiency of anti-oxidant strategies [64]. Here, pioneering results have been presented, which may open up the development of such a tool. They show that the catalysis of electrochemical ORR by cells can be easily characterized by a fast electrochemical measure such as CV. The electroanalytical device used here (Fig 1) could be implemented with cells adhered or deposited on the electrode

surface. The catalysis of the ORR induced by the cells could be quantified by CV, as done here, following a standardised protocol. For the future, more sophisticated electroanalytical methods can also be envisaged, such as staircase voltammetry or differential pulse voltammetry, which generally increase the sensitivity of the measurements and can help to quantify the effect of the cells on ORR more accurately with respect to the controls achieved without cells. The method could allow very fast assessment of the efficiency of the cell protective systems. The impact of toxic agents and antioxidant strategies on the cell protective systems could also be quantified by performing, in parallel, several assays in the presence or absence of the compounds under investigation. A multiple-well electroanalytical device would be fully appropriate for this purpose.

Obviously, only preliminary leads have been discussed here relating to the possible electron transfer mechanisms. Basic studies are now required to unravel the mechanisms of electrochemical ORR catalysis by cells and to enable the link to be made between the electrochemical response and the oxidative protective system. A new wide field of investigation is opened up.

## Conclusions

For the first time, animal and human cells have been shown to catalyse the electrochemical reduction of oxygen. The procedure detected differences in the behaviour of the two cell lineages that were checked. Different observations of, and the literature available on, the same catalysis observed with microbial cells suggest the involvement of antioxidant-related proteins. These pioneering results may be the basis of a bioanalytical procedure for characterizing the oxidative protective system of cells and their reactions to external agents.

## Acknowledgments

The authors thank Haouria Belkhefa (Fonderephar) for helpful advice on cell cultures, L. Etchevery (Laboratoire de Génie Chimique) for his help with electrochemical set-ups, and Dr Leila Haddioui (Fonderephar) for providing cells.

## Author Contributions

**Conceptualization:** Alain Bergel.

**Funding acquisition:** Alain Bergel.

**Investigation:** Simon Guette-Marquet.

**Methodology:** Simon Guette-Marquet, Christine Roques, Alain Bergel.

**Project administration:** Christine Roques, Alain Bergel.

**Supervision:** Christine Roques, Alain Bergel.

**Validation:** Simon Guette-Marquet, Christine Roques, Alain Bergel.

**Visualization:** Simon Guette-Marquet, Alain Bergel.

**Writing – original draft:** Simon Guette-Marquet, Alain Bergel.

**Writing – review & editing:** Simon Guette-Marquet, Alain Bergel.

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
