## [Decision Letter · Decision Letter 0]

10 Mar 2021

PONE-D-21-00974

Catalysis of the electrochemical oxygen reduction reaction (ORR) by animal and human cells

PLOS ONE

Dear Dr. Bergel,

Thank you for submitting your manuscript to PLOS ONE. After careful consideration, we feel that it has merit but does not fully meet PLOS ONE’s publication criteria as it currently stands. Therefore, we invite you to submit a revised version of the manuscript that addresses the points raised during the review process.

We look forward to receiving your revised manuscript.

Kind regards,

Zafar Khan Ghouri

Academic Editor

PLOS ONE

Journal Requirements:

Additional Editor Comments (if provided):

Dear Author/s,

The reviewer has commented on PONE-D-21-00974 and indicated that it is not acceptable for publication in its present form.

However, if you feel that you can suitably address the reviewer's comments (included below), I invite you to revise and resubmit your manuscript.

Please carefully address the issues raised in the comments.

Zafar Khan Ghouri, Ph.D.

Reviewers' comments:

Reviewer's Responses to Questions

**Comments to the Author**

1. Is the manuscript technically sound, and do the data support the conclusions?

Reviewer #1: Yes

2. Has the statistical analysis been performed appropriately and rigorously? 

Reviewer #1: Yes

3. Have the authors made all data underlying the findings in their manuscript fully available?

Reviewer #1: Yes

4. Is the manuscript presented in an intelligible fashion and written in standard English?

Reviewer #1: Yes

5. Review Comments to the Author

Reviewer #1: The manuscript titled Catalysis of electrochemical oxygen reduction reaction (ORR) by animal and human cells presents the findings of animal and human cells from the Vero and MRC5 lineage, respectively, being able to induce ORR after some incubation time. The manuscript provides fresh insights into a niche field that may yield some promise in the future if similar work is concentrated on it. The posed application of biosensors capable of characterizing protective systems of cells against oxidative stress is very interesting, and the presented work introduces it softly. However, more conclusive undertakings on how this work may concretely lead to that is lacking.

1. Build on the sentence in line 47: “Two different types of microbial ORR catalysis can be distinguished” – it seems incomplete.

2. Grammatic errors and incorrect writing styles are present throughout the manuscript. For example (line 75): ..(glycolysis, fermentation…) and (line 136): Cyclic Voltammetry (CV) used a VMP-3 potentiostat … and (line 140): connect the sentence from line 140 to sentence in 141 to be under the same paragraph.

3. Under section “Electrode and electrochemical reactors” add an equation(s) to clearly show how you go from your reference electrode potential to SHE (or RHE).

4. Under the same section, consider adding a testing schematic. This is particularly important since this is a novel work and it would be important to clearly convey without any ambiguity how testing was performed.

5. In line 115, elaborate more why PDL coating was done on some of the electrodes.

6. In line 137/138, state clearly that the differences between the CV cycles was in terms of current response, and not a difference in the method (scan rate or scan range for example) of taking those CV curves.

7. There’s a contradiction between pH values after the 72-hour of incubation in lines 253 and 255. Is the pH value after 72-hours of incubation equal to 6.8 or 6.7?

8. For all polarization curves, the x-axis is labelled (Potential/ref(V)) which is very vague for anyone reading this work. It is recommended that all polarization data be plotted with respect to SHE or RHE for consistency and comparison with literature and future works that use SHE and RHE as standard potentials during discussion.

9. Fix formatting of Figure 4 (i.e., borders). Make sure to have all figures formatted the same (i.e., either bold or un-bold the text of axis titles, maintain same font, etc.).

6. PLOS authors have the option to publish the peer review history of their article (what does this mean?). If published, this will include your full peer review and any attached files.

Reviewer #1: No

---

## [Author Response · Author response to Decision Letter 0]

19 Apr 2021

Reviewer #1: The manuscript titled Catalysis of electrochemical oxygen reduction reaction (ORR) by animal and human cells presents the findings of animal and human cells from the Vero and MRC5 lineage, respectively, being able to induce ORR after some incubation time. The manuscript provides fresh insights into a niche field that may yield some promise in the future if similar work is concentrated on it. The posed application of biosensors capable of characterizing protective systems of cells against oxidative stress is very interesting, and the presented work introduces it softly. However, more conclusive undertakings on how this work may concretely lead to that is lacking.

Reply: Thank you very much for the nice comments. We particularly appreciate the hopes you express for the possible development that this work could open up. 

In accordance with the last remark, a paragraph has been added at the end of the Discussion section in order to better outline the idea we have of the analytical device that could be developed on the basis of this work. By the way, the addition of the scheme of the experimental setup (requested by the reviewer in point 4) will also be useful here: 

“The electroanalytical device used here (Figure 1) could be implemented with cells adhered or deposited on the electrode surface. The catalysis of the ORR induced by the cells could be quantified by CV, as done here, following a standardised protocol. For the future, more sophisticated electroanalytical methods can also be envisaged, such as staircase voltammetry or differential pulse voltammetry, which generally increase the sensitivity of the measurements and can help to quantify the effect of the cells on ORR more accurately with respect to the controls achieved without cells. The method could allow very fast assessment of the efficiency of the cell protective systems. The impact of toxic agents and antioxidant strategies on the cell protective systems could also be quantified by performing, in parallel, several assays in the presence or absence of the compounds under investigation. A multiple-well electroanalytical device would be fully appropriate for this purpose.” 

1. Build on the sentence in line 47: “Two different types of microbial ORR catalysis can be distinguished” – it seems incomplete.

Reply: This sentence intended to introduce the following two paragraphs. To be clearer it has been completed to: “Two different types of microbial ORR catalysis can be distinguished, as detailed below”.

2. Grammatic errors and incorrect writing styles are present throughout the manuscript. For example (line 75): ..(glycolysis, fermentation…) and (line 136): Cyclic Voltammetry (CV) used a VMP-3 potentiostat … and (line 140): connect the sentence from line 140 to sentence in 141 to be under the same paragraph.

Reply: line 75, the sentence has been modified to: “Various metabolic pathways, such as glycolysis and fermentation, and including aerobic respiration, have been considered as the source of the electrons transferred to the anode.” 

line 136, the sentence has been modified to: “Cyclic voltammograms (CVs) were recorded using a VMP-3 potentiostat”

line 140 has been connected to line 141

3. Under section “Electrode and electrochemical reactors” add an equation(s) to clearly show how you go from your reference electrode potential to SHE (or RHE).

Reply: A sentence has been added to clarify the correspondence: “The potentials can be expressed relative to the standard hydrogen electrode (SHE) by adding 0.290 mV to the values given in the text.”

4. Under the same section, consider adding a testing schematic. This is particularly important since this is a novel work and it would be important to clearly convey without any ambiguity how testing was performed.

Reply: A schematic has been added (Figure 1). By the way, this figure can also be useful to represent the possible future analytical device, according to the first comment of the reviewer. 

5. In line 115, elaborate more why PDL coating was done on some of the electrodes.

Reply: A sentence and a reference have been added, not in line 115, which was in the Mat and Meth section, but in the Result section (lines 301 to 305): “In some cases the electrodes were pre-treated with poly-D-lysine (PDL) with the intention of promoting cell adhesion on their surface. PDL is a synthetic, positively charged polymer, which binds to the negatively charged cell membrane through electrostatic interaction and is thus commonly used to promote cell adhesion on solid surfaces [47].” 

6. In line 137/138, state clearly that the differences between the CV cycles was in terms of current response, and not a difference in the method (scan rate or scan range for example) of taking those CV curves.

Reply: A sentence has been added: “In each experimental set, several reactors were run in parallel, always with one or two control reactors without cells. The values of all the electroanalytical parameters were constant.” 

7. There’s a contradiction between pH values after the 72-hour of incubation in lines 253 and 255. Is the pH value after 72-hours of incubation equal to 6.8 or 6.7?

Reply: After 72 h incubation, the pH was slightly acidified from 7.2 to 6.8, but we showed that this acidification had no significant effect with a solution at pH 6.7 in order to be sure (small security margin). The sentence has been clarified to: “The possible impact of such an acidification on ORR was assessed by recording control CVs in the complete growth medium without cells after decreasing its pH from 7.2 to 6.7 by adding a small quantity of hydrochloric acid”

8. For all polarization curves, the x-axis is labelled (Potential/ref(V)) which is very vague for anyone reading this work. It is recommended that all polarization data be plotted with respect to SHE or RHE for consistency and comparison with literature and future works that use SHE and RHE as standard potentials during discussion.

Reply: If the CV were plotted with respect to SHE (or another reference), the potential scan limits would look strange, with decimal values. Consequently, to make the CVs clearer, we have specified on each X-axis label that the DropSens reference was used and we have recalled in each legend that the potentials with respect to SHE can be obtained by adding 290 mV to the present values.

9. Fix formatting of Figure 4 (i.e., borders). Make sure to have all figures formatted the same (i.e., either bold or un-bold the text of axis titles, maintain same font, etc.).

Reply: Figure 4 has been put to the same format and all other figures have been carefully checked to be in the same format.

---

## [Editor Report · Decision Letter 1]

23 Apr 2021

Catalysis of the electrochemical oxygen reduction reaction (ORR) by animal and human cells

PONE-D-21-00974R1

Dear Dr. Bergel,

We’re pleased to inform you that your manuscript has been judged scientifically suitable for publication and will be formally accepted for publication once it meets all outstanding technical requirements.

Kind regards,

Zafar Khan Ghouri

Academic Editor

PLOS ONE

---

## [Editor Report · Acceptance letter]

27 Apr 2021

PONE-D-21-00974R1 

Catalysis of the electrochemical oxygen reduction reaction (ORR) by animal and human cells 

Dear Dr. Bergel:

I'm pleased to inform you that your manuscript has been deemed suitable for publication in PLOS ONE. Congratulations! Your manuscript is now with our production department. 

Kind regards, 

on behalf of

Dr. Zafar Ghouri 

Academic Editor

PLOS ONE